# Control Strategies for Daylight and Artificial Lighting in Office Buildings—A Bibliometrically Assisted Review

Daniel Plörer [1], Sascha Hammes [1,2], Martin Hauer [1,2], Vincent van Karsbergen [1] and Rainer Pfluger [1,*]

1   Unit of Energy Efficient Building, University of Innsbruck, A-6020 Innsbruck, Austria;
    daniel@optifas.eu (D.P.); sascha.hammes@bartenbach.com (S.H.); martin.hauer@bartenbach.com (M.H.);
    Vincent.van-Karsbergen@uibk.ac.at (V.v.K.)
2   Bartenbach GmbH, A-6071 Aldrans, Austria
*   Correspondence: rainer.pfluger@uibk.ac.at; Tel.: +43-512-507-63602

**Abstract:** A significant proportion of the total energy consumption in office buildings is attributable to lighting. Enhancements in energy efficiency are currently achieved through strategies to reduce artificial lighting by intelligent daylight utilization. Control strategies in the field of daylighting and artificial lighting are mostly rule-based and focus either on comfort aspects or energy objectives. This paper aims to provide an overview of published scientific literature on enhanced control strategies, in which new control approaches are critically analysed regarding the fulfilment of energy efficiency targets and comfort criteria simultaneously. For this purpose, subject-specific review articles from the period between 2015 and 2020 and their research sources from as far back as 1978 are analysed. Results show clearly that building controls increasingly need to address multiple trades to achieve a maximum improvement in user comfort and energy efficiency. User acceptance can be highlighted as a decisive factor in achieving targeted system efficiencies, which are highly determined by the ability of active user interaction in the automatic control system. The future trend is moving towards decentralized control concepts including appropriate occupancy detection and space zoning. Simulation-based controls and learning systems are identified as appropriate methods that can play a decisive role in reducing building energy demand through integral control concepts.

**Keywords:** office buildings; control strategies; daylight; artificial lighting; energy efficiency; comfort; user-centered systems

## 1. Introduction

Buildings are responsible for more than one third of the world's energy demand [1,2]. For this reason, national governments and international organizations worldwide are paying special attention to the building sector while defining new energy policies and promoting the introduction of more sustainable and energy-efficient solutions in buildings. On the one hand, requirements have been defined for the design of new high-performance buildings that use nearly zero energy. On the other hand, action plans for the renovation of the existing building stock are proposed. In this context, Economidou et al. provide an overview of the development of energy efficiency policies on new buildings and building renovations in the European Union and the related instruments to promote these measures [3]. It identifies progress and challenges as well as potentials that could promote energy savings in buildings. Much of the energy demand in buildings is currently due to outdated building practices, inefficient systems and equipment, and inadequate technical control systems [3]. Improved control strategies therefore play an important role in achieving these goals. Several studies show an energy savings potential of integrated daylight and artificial lighting controls to be 30–80% in artificial lighting energy [4–6] and 3–43% savings in heating, ventilation, and air conditioning (HVAC) energy demand [7].

Integrated control strategies for blinds and artificial light can be divided into closed-loop and open-loop control systems. While in a closed-loop the control system reacts to a

sensor feedback (e.g., interior lighting sensors for artificial lighting control), in an open-loop approach, sensor information is mostly processed by an advanced simulation-based control system, which then provides set values for actors. In this way, the contradicting requirements on a façade, such as daylighting, solar gains control, avoiding discomfort glare and enhancing visibility to the outside, can be optimally balanced with the operation of the artificial lighting system or HVAC components. In this context, Jain and Garg published a comprehensive review paper on the state of the art of control strategies for window blinds [8].

However, Konstantoglou et al. evaluate in their review that the user has a decisive influence on the effectiveness of these control strategies, so that, in addition to conceptual difficulties, installation and configuration problems, the user can be named as a major cause for the wide range of estimates of energy savings in control strategies [9]. Often, the user takes measures to create comfort that negate the effects of energy-saving measures such as is the case with automated blinds [10,11]. Excessive and uncontrolled daylight can lead to visual discomfort due to glare and high cooling loads in the summer period. Users usually close manually operable blinds to reduce visual discomfort or excessive sunlight [12]. Most of the time they remain in this state for the rest of the day, although without blinds there would be no more complaints [13]. Instead, artificial light is switched on during the day and increases electrical energy demand and internal loads. Furthermore, it reduces visibility to the outside and decreases user satisfaction in the workplace [14]. User interventions can therefore have a negative impact on energy demand. Current control strategies are usually designed to either operate under an energy efficiency objective or to increase user comfort. In this context, this work aims to answer the following research questions (RQ) via a thorough review of literature:

(RQ1)  How can the mutually influencing criteria user comfort and energy efficiency be optimized simultaneously in lighting control strategies in office buildings?
(RQ2)  What are the strengths and weaknesses of systems and concepts applied?
(RQ3)  How are future requirements for control systems addressed by current research trends?

In doing so, current obstacles of existing control approaches and resulting research gaps will be identified for future investigations in this area. The identification of suitable control strategies that combine both objectives will be carried out in the course of a critical literature analysis as well as a trend analysis based on bibliometric studies. The investigations will focus on the control systems as well as on the required sensor technology and system architecture. As daylighting systems are mainly applied in office buildings, this will be the focus edifice type of this review.

## 2. Materials and Methods

### 2.1. Literature Research

In order to give an overview of the topic, a comprehensive literature research was performed. The search engines of SAGE journals, Taylor and Francis, Wiley, Emerald Publishing, the International Building Performance Simulation Association (IBPSA), Web of Science, Elsevier (Science Direct), Springer, Inderscience, and Google Scholar were used to cover all relevant publisher data bases. Care was taken to use the publisher's own search tools as well as independent search engines from third parties. First, a set of keywords, which correlates to the research question, was formulated. Seven main keywords were defined and for each a set of synonyms was formulated. Table 1 shows the initial keywords including synonyms.

**Table 1.** List of main key words and corresponding synonyms as used in the literature search.

| Main Keyword | Review | Building | Daylight | Lighting | Individual | Control | Energy Efficiency |
|---|---|---|---|---|---|---|---|
| Synonyms | state of the art | buildings | blind | light | comfort | controlling | energy savings |
| | | office | blinds | | user-centered | building automation | energy reduction |
| | | offices | facade | | user | strategies | building performance optimization |
| | | | envelope | | personalized | strategy | |
| | | | shading | | | systems | |
| | | | slats | | | system | |
| | | | | | | MPC | |

From the defined keywords, the search string could be derived. For this purpose, each main keyword was logically-OR linked with its associated synonyms (Table 1) and all resulting keyword-synonym-strings were then logically-AND linked resulting in the complete search string. The search engines listed in Table 2 were used for the literature search. The search was directed exclusively to designated review papers. To ensure adequate timelines, the review paper search was limited to the period from 2015 to 2020. The first literature search was conducted in August 2020 and, for the purpose of completing 2020, an update was performed in January 2021 for all search engines. The created search string was applied to the entire article if possible (full text search). To further limit the selection, an additional search string was created for the filter criteria title and keywords, which are offered by some search engines. Since some search engines have limitations, such as the maximum number of Boolean links (max. 8 per field, ScienceDirect by Elsevier), the length of the URL, i.e., the length of the web address (Google Scholar) or a missing function of the full text search (Web of Science), the search string had to be adjusted accordingly. Each modification of the search string was formed based on a subset of the initial keyword-set (Table 1). In this way, the expected set of search results forms a super-set to the set of results one would get, when the entire keyword-set would be used. For creating the sub-strings, primarily keywords and synonyms from the columns 'building', 'lighting', and 'control' of Table 1 were used. These three main keywords were logically-OR linked with their corresponding synonyms and the three resulting strings were logically-AND linked. The required substrings for keywords and titles were formed equally. When the reduction of the number of synonyms was necessary, care was taken to use only those synonyms that produced the highest number of results of all the search string combinations. In summary, a literature search was conducted with filtering by keywords, year of publication, and review paper labeling, with the latter filtering primarily done manually. All search services and the applied search methodologies are listed in Table 2.

Table 2 lists the search methodology and search results for all literature databases used. The list of review results is derived from the searches according to the order shown in Table 2, i.e., when excluding overlaps, the hits listed first in Table 2 were used first and subsequent ones were declared duplicates. If an already identified hit was found with a search engine, it was listed under used with the search engine that is listed first on the order of Table 2 and listed as duplicate in brackets if it was found in further search engines. Since publisher-owned search engines are listed first, identified duplicates preferentially fall to independent third-party search engines. The third party search engines Web of Science and Google Scholar consequently served as supplementary sources. For the initial selection of potentially useful review articles, titles and abstracts of all hits from the literature search (Table 2) were reviewed and identified. These were then used for a more detailed review (passed to the detailed search, Table 2). In the detailed search, the prefiltered articles were evaluated to determine if they could contribute to answering the research questions. This step was necessary to avoid biasing later statistical analyses by misattributed publications. All articles whose content was of interest were evaluated in detail and used in the study. These articles are referred to below as review articles (used publications, Table 2) because

the initial search criteria limited the search results to these types of publications. A total of 41 review articles were used for the evaluation after manual screening. References of the review articles that are in line with the research question of this paper were extracted and further utilized to perform the evaluation of the current trends in the scientific literature in the field of daylighting and artificial lighting control strategies. Those articles are in the following referred to as research articles, of which a total of 575 were selected.

**Table 2.** List of main research databases used to perform the literature search (* Complete search string, ** Substrings); The data in the columns 'Transfer to detailed search' and 'Used publications' are without duplicates and with duplicates, the latter value is listed in brackets.

| Database/Search Engine | Search Methodology | Filter | Hits | Transfer to Detail Search (with Duplicates) | Used Publications (with Duplicates) |
|---|---|---|---|---|---|
| SAGE journals * | (Anywhere AND Title) OR (Anywhere AND Keywords) | Journals | 53 | 4 | 0 |
| Taylor & Francis * | (Anywhere AND Title) OR (Anywhere AND Keywords) | Journals | 82 | 5 | 3 |
| Wiley * | (Anywhere AND Title) OR (Anywhere AND Keywords) | Journals | 111 | 1 | 1 |
| Inderscience Publishers * | Anywhere AND Title | Article | 11 | 0 | 0 |
| Emerald Publishing ** | Anywhere AND Title | Article | 81 | 0 | 0 |
| Elsevier (Science Direct) ** | Anywhere AND (Title OR Abstract OR Keywords) | Review-Article | 427 | 53 | 27 |
| IBPSA ** | Anywhere | Article AND Discipline: Energy & Engineering | >1000 | 0 | 0 |
| Springer ** | Title | Article | 400 | 2 | 0 |
| Web of Science ** all databases | Title | Journals | 211 | 7 (15) | 1 (7) |
| Google Scholar ** | Title | Article | 160 | 22 (33) | 9 (18) |
| **Total** | - | - | **>2536** | **94 (113)** | **41 (56)** |

## 2.2. Evaluation Methodology

The trends in day- and artificial lighting control strategies were studied by elaborating the evolution over time of the correlation strength between relevant terms found in titles, abstracts, and keywords of the research articles. An important reference point of the underlying timeline was chosen at 2010 because, around that time, technological innovations concerning lighting and building control systems became more widely spread. The change from fluorescent tubes to LED allowed a large increase in energy efficiency of lighting systems in the partial load range, which made dimming much more effective for energy saving. Noticeable increased application of Information and communications technology (ICT) facilitated the implementation of advanced sensor- and control systems. This technological shift also promoted the development of complex control strategies focusing on multiple targets. For the trend analysis, we separated the selected research articles published before this innovation shift around 2010 from articles published afterwards. To enhance the accuracy of the trend analysis and to enable investigation of new trends, which

have not been widely discussed before 2010, an additional separation was made for articles published before and after 2015. The amount of research articles found in each time period is listed in Table 3. For each period, the same methodology was applied.

**Table 3.** Classification of the filtered research articles for the three time periods.

| Period | Period 1 | Period 2 | Period 3 |
|---|---|---|---|
| Year range | 2010 and earlier | 2011–2015 | 2016–2020 |
| Number of research articles | 160 | 260 | 155 |

The basic set of keywords is found by collecting all keywords provided by the authors of each research paper published in the particular time period. Since the formulation of the keywords can differ significantly per author, a new set of terms was defined, whereby every term corresponds to a category of keywords. Hereafter, the elements in this set of terms are referred to as 'items', following the VOSviewer nomenclature [15]. In that sense, the analysis of the correlation strength was then performed between items, as described in the following.

In order to visualise the interconnection between the items, the VOSviewer tool (version 1.6.16) [15] was used. Following the same methodology as used in VOSviewer, a similarity measure was employed to measure the correlation strength between two items. Van Eck et al. give an overview over the most popular similarity measures found in scientometric research and provide a generalized similarity index (Equation (1)), from which the discussed similarity measures can be derived [16]. A detailed explanation of different similarity measures was provided by Warrens and colleagues [17]. For this research, we first define an occurrence matrix $O$, similar to the one described in [16], with $N$ columns and $M$ lines, whereby $N$ represents the number of the investigated research articles and $M$ the number of found items. Each entry of $O$, where the column, corresponding to a certain item, meets the line, corresponding to a certain article, is either of value one if at least one keyword of this article appears in the keywords-set corresponding to the item, or zero otherwise. After performing the matrix multiplication between the transposed matrix $O^T$ and the original matrix $O$, we obtain the squared matrix $C$ of dimension $M \times M$, which denotes the occurrences resp. co-occurrences of the keywords in all the investigated articles. Both lines and columns of matrix $C$ refer to the items. The elements in the main diagonal of the matrix $C$ represent the total occurrence of each item, which is the number of research articles, in which at least one of the keywords out of the keyword-set assigned to the specific item is listed. Every other element of $C$ represents the co-occurrence of the respective item pair, which is the number of research articles, in which at least one keyword out of each of the two keyword-sets assigned to the two items appears. As Van Eck and colleagues describe, the most popular similarity measures can be derived directly from the matrix $C$ [16]. The authors provide the generalized similarity index as a function of the co-occurrence $c_{ij}$, the occurrences of the two observed items $c_{ii}$, $c_{jj}$ (whereby $i$ and $j$ stand for different items) and a parameter $p$ in the range of $\mathbb{R} \setminus 0$, which specifies the form of the similarity index

$$S(c_{ij}, c_{ii}, c_{jj}, p) = 2^{\frac{1}{p}} c_{ij} \left( c_{ii}^p + c_{jj}^p \right)^{-\frac{1}{p}}. \tag{1}$$

For the study at hand, the joint conditional probability measure (*jcpm*) is used, which derives from the general similarity index by setting the parameter $p$ to $-1$:

$$jcpm(c_{ij}, c_{ii}, c_{jj}) = S(c_{ij}, c_{ii}, c_{jj}, -1) = \frac{1}{2} \left( \frac{c_{ij}}{c_{ii}} + \frac{c_{ij}}{c_{jj}} \right). \tag{2}$$

This measure is also used by McCain for investigating the structure of biotechnology R&D [18]. For all three observed time periods, the co-occurrence matrix $C$ and consequently the *jcpm* has been evaluated. By comparing the *jcpm* of a certain item-pair of the three

time periods, it can be observed whether the correlation of the two items is increasing or decreasing. The evaluation of such trends is presented in the following section for the most relevant item pairs.

## 3. Results

### 3.1. Results of the Multilevel Analysis (Review- and Research-Level)

Before the numerical results of the VOSviewer output are presented from the next subsection onward, an evaluation of the generated network (Figure 1) should take place on a descriptive level. The network representation shows a high degree of interconnectedness between all items ($\varnothing = 19.63 \pm 5.84$ links per item), but with a strong variation of the occurrences (range 192 and an average of 36). Furthermore, the items 'building energy savings' and 'lighting control' form the largest focal points, followed by the three control strategies 'learning system', 'predictive control' and 'rule based'. Due to the strong interconnectedness of all items for 'building energy savings', this item is positioned relatively central in the network representation.

The occurrences of the items can be considered as a measure to evaluate the relevance in relation to the underlying data set [15,19]. In order to enable a comparison over different observation periods, the occurrences were normalized in relation to the reference quantity of the articles in the respective time period. It can be seen that the emerging foci of the entire VOSviewer based analysis (Figure 1) correspond to the main terms of the keywords of the literature search (see also Table 1). Among others, the main keywords 'energy', 'control', and 'lighting' correspond exactly to the largest occurrences of the VOSviewer based analysis (normalized occurrences of building energy savings: 0.3443, lighting control: 0.1652, time period: All). Accordingly, there is consistency between the results of the literature review analysis and those research articles that were used for the trend analysis.

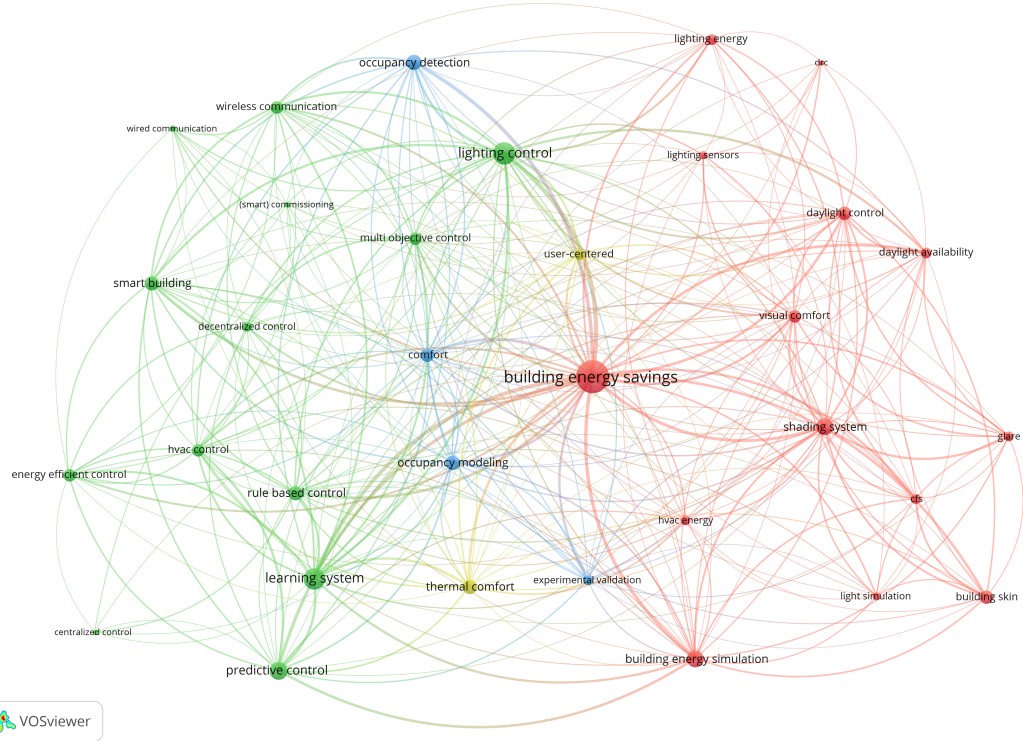

**Figure 1.** Network analysis of the identified research articles, classified according to the set items, visualized via VOSviewer.

Based on the investigated review papers, the following research foci can be identified and classified (Table 4). Here, too, the greatest focus of interest can be found in the area of energy efficiency.

**Table 4.** Identification and classification of the filtered review articles.

| Review Focus | Associated Review Papers |
|---|---|
| System acceptance and usability | [9,20–24] |
| Energy demand and the degree of automation | [8,9,20,21,25–34] |
| Energy demand and user behaviour | [8,9,11,20–22,24–26,28,29,31,32,35–37] |
| Smart commissioning | [8,22,23,25,30,32] |
| Use of multi-objective controls and identification of the need for them | [8,20,23,25,27,30,33,34,38–43] |
| System architecture and design of the control logic | [8,11,20,22,26,27,31,36,37,41,44] |
| Influences of sensor resolution and positioning and lighting zoning on energy demand | [8,9,11,22,23,26,27,30,31,33,35,36,41,45–47] |

The results of the review-level literature search are merged below with the results of the VOSviewer based analysis to answer the research question, namely how can energy efficiency and user comfort be addressed in lighting control strategies in office buildings, what are the strengths and weaknesses of systems and concepts applied, and how are future requirements for control systems addressed by current research trends.

### 3.2. The Relevance of Multi-Objective Control

The bibliometric investigations on the level of the research articles show that the general interest in building energy demand as well as in comfort criteria, especially visual comfort, grows significantly over the periods considered. This development is also represented in the percentage increase $(1 + p)$ of the normalized occurrences from time period 1 to time period 3. For the item building energy savings, this results in an increase of interest of 95.8% (see also Table 5). For visual comfort, the increase is 188.5%. The same applies to the user-centeredness of systems (increase in relevance by 64.9%).

**Table 5.** Normalized occurrences of different target criteria of a building control as well as the normalized occurrences of the items lighting control and multi-objective control, determined by using VOSviewer.

| Item | Normalized Occurrences | | | |
|---|---|---|---|---|
| | 2010 and Earlier | 2011–2015 | 2016–2020 | All |
| building energy savings | 0.2438 | 0.3269 | 0.4774 | 0.3443 |
| comfort | 0.0688 | 0.0500 | 0.0516 | 0.0557 |
| visual comfort | 0.0313 | 0.0385 | 0.0903 | 0.0504 |
| thermal comfort | 0.0500 | 0.0654 | 0.0839 | 0.0661 |
| glare | 0.0125 | 0.0423 | 0.0323 | 0.0313 |
| user-centered | 0.0313 | 0.0308 | 0.0516 | 0.0365 |
| lighting control | 0.1688 | 0.1462 | 0.1935 | 0.1652 |
| multi-objective control | 0.0000 | 0.0577 | 0.0839 | 0.0487 |

The literature reviewed allows a separation into multi-objective controls to optimise daylight, artificial lighting and energy performance [25,27,33,42,43]—and single-objective controls [8,20,23,27,30,34,38–40] that focus on user comfort or energy efficiency. Control systems with the objective of fulfilling several target criteria seem to be gaining in importance. In the literature review of Ding, Yu, and Si, the development of control systems in building technology is examined for scientific literature published from 1993 to 2017, focusing on both energy savings and visual comfort [42]. The results list an increasing research interest in visual comfort. The dominant trend of research and application is therefore identified as moving from energy savings to a combination of energy saving and visual comfort, creating a more human-centered office lighting. Tabadkani et al. review simulation-based studies using automatic shading control methods balancing human comfort and energy savings [38]. For the period reviewed, building controls with one target criteria were identified as more widespread, with a trend towards the development of

multi-target controls. In this respect, it can be concluded that control systems with the objective of meeting multiple target criteria are gaining importance.

The increasing interest in combining different target criteria in control strategies is also reflected in the growing importance of multi-objective control systems in the VOSviewer based analysis. Regarding the research articles on which the evaluation is based, there is no relevant interest in multi-objective control systems in the time period before 2010 whose objectives are based on energy criteria as well as on user comfort criteria. The occurrence of multi-objective control systems positions itself directly to a level of 0.0577 in the following observation period (see also Table 5). This corresponds to an occurrence of 27.1% above the arithmetic mean of the other target criteria excl. the item building energy savings (Table 5, period: 2011–2015). Looking at the most recent literature (period 3), the trend in interest in multi-objective controls (normalized occurrences) shows a further increase of 45.4% compared to the previous period (Table 5).

In order to evaluate the relationships of individual terms to each other, the VOSviewer represents the value of the linkage by means of the joint conditional probability measure (*jcpm*, calculation Formula (2)). By means of this normalized valuation, the target criteria of the control system can be related to the artificial lighting trade and assessed quantitatively. Above all, the linkages are most pronounced in the published literature of period 3. This concerns especially the linkages of lighting control with energetic aspects (*jcpm*: 0.3514) as well as those of lighting control to user-centeredness (*jcpm*: 0.2375) and to visual comfort (*jcpm*: 0.2095). Thermal comfort and artificial lighting control show no correlation in the present dataset (Table 6). For thermal comfort, the VOSviewer based analysis identifies only a direct relationship to daylight-based control logics as well as HVAC systems. This can be attributed to the fact that the influence of the HVAC and daylight trades on thermal aspects are weighted higher than that of artificial light.

**Table 6.** Joint conditional probability measure (*jcpm*) of the items lighting control and multi-objective control networked with different target criteria of a building control, determined by using VOSviewer.

| Item | Lighting Control (*jcpm*) | | | | Multi-Objective Control (*jcpm*) | | | |
|---|---|---|---|---|---|---|---|---|
| | 2010 and Earlier | 2011–2015 | 2016–2020 | All | 2010 and Earlier | 2011–2015 | 2016–2020 | All |
| building energy savings | 0.3447 | 0.2475 | 0.3514 | 0.3037 | 0.0000 | 0.1569 | 0.2261 | 0.1834 |
| comfort | 0.1279 | 0.0516 | 0.1583 | 0.1044 | 0.0000 | 0.0718 | 0.1010 | 0.0670 |
| visual comfort | 0.1185 | 0.0632 | 0.2095 | 0.1350 | 0.0000 | 0.0833 | 0.0742 | 0.0702 |
| thermal comfort | 0.0000 | 0.0000 | 0.0000 | 0.0000 | 0.0000 | 0.1255 | 0.0000 | 0.0620 |
| glare | 0.0000 | 0.0586 | 0.1167 | 0.0661 | 0.0000 | 0.0788 | 0.0000 | 0.0456 |
| user-centered | 0.2370 | 0.2270 | 0.2375 | 0.2326 | 0.0000 | 0.0000 | 0.1010 | 0.0417 |
| multi-objective control | 0.0000 | 0.1860 | 0.1103 | 0.1387 | | | | |

From the presentation form of Table 6, the single target criteria (i.e., visual comfort, energy efficiency) could be set in relation to the item lighting control systems and their dependencies could be measured via the *jcpm*. However, it cannot be read out that these are control systems addressing several target criteria or control systems serving only single targets. To realize such a consideration, the *jcpm* of the single objective criteria is set in relation to the item multi-objective control (Table 6). With this comparison, a demonstrable increase can be recorded between the items' building energy demand and multi-objective control (increase in *jcpm* of 44.1% between time period 2 and time period 3, Table 6). Likewise, the correlation of comfort and user-centeredness to the item multi-objective control has gained importance mostly in the period of 2015–2020 in relative as well as absolute terms, as can be seen from the occurrences of the items (see also Table 5).

### 3.3. User Acceptance and Its Influence on System Performance

The importance of considering energy aspects as well as comfort criteria is due to their mutual influences. Energy targets and user comfort can often conflict with each other [39]: Bellia, Fragliasso, and Sefanizzi therefore identified user acceptance and satisfaction, which can be seen as the result of fulfilled user comfort criteria, as an important boundary for the performance of daylight-dependent lighting and shading controls [22]. If this acceptance is not guaranteed, users tend to interfere with automatic controls and even deactivate them. Consequently, the efficiency and functionality of a system is fundamentally affected [22]. Other review papers also identify user acceptance as a crucial factor for achieving the desired system efficiency, which is significantly related to the functioning of the system, ensuring intervention options in the building control system and intuitive usability [9,20–23,32,48,49]. Although complex control strategies have the potential to significantly save energy, as noted by Konstantoglou and Tsangrassoulis in their literature review, their level of complexity has an impact on efficiency due to limited user acceptance and can therefore lead to limitations in energy performance [9]. In order to reduce malfunctions and the complexity of installation and operation, smart commissioning measures can contribute [8,22,25].

Fiorito et al. conclude from their literature review that a major aspect for inefficient use of energy in buildings is the mismatch of the interaction between building systems and occupants [24]. Konstantoglou and Tsangrassoulis state in their review that occupants often adjust manually operated shading systems to their personal preferences [9]. In this context, Tabadkani et al. list that most studies focus on automatic shading controls but include little to no strategy to map preferences of individual users in a shared work environment [38]. Since energy savings are largely determined by the degree of automation of the control system and system performance depends on user behaviour or acceptance [8,9,11,20–22,24–37], the personal preferences and requirements of users should be taken more into account when optimising control algorithms to achieve better system performance and thus higher energy savings [23,38]. Mofidi and Akbari also therefore recommend energy management systems that combine both criteria [39].

### 3.4. The Design of the Control Logic

According to Salimi and Hammad, current control practices achieve little to no reasonable trade-off between minimizing energy costs and maximizing comfort and satisfaction [36]. Boodi et al. also recognize the need for further research to find an appropriate trade-off between energy and comfort requirements [34]. For this reason, this thesis investigated the question of which systems offer the potential to combine user comfort and energy efficiency in a targeted manner.

For the investigations of the relevant control logics, a differentiation was made between rule-based systems, predictive controls, and learning systems. While the importance of rule-based systems based on the VOSviewer based analysis remains almost unchanged over the years (range: 11.9%), learning systems, on the other hand, are becoming increasingly important (increase in normalized occurrences: 220%, time period 1 to period 3). This is also true with respect to their relationship to trade-specific systems (see also Table 7). Predictive systems took a high importance especially in the time period 2011 to 2015 (normalized occurrences: 0.1615, Table 8) and this especially in the HVAC domain (*jcpm*: 0.351, Table 7). In addition, the VOSviewer based analysis lists that predictive systems record little to no consideration of daylighting, and the dependencies of this item on the other trades also decreases in the more recent contribution periods (see also *jcpm* in Table 7). The increasing linkage of learning systems to multi-objective control (increase in *jcpm* from time period 2 to period 3: 138.2%, time period 1 having no *jcpm*) can be attributed to the fact that learning systems can be used to solve more complex problems.

Since user behavior has a strong influence on the energy consumption of buildings, their integration into control logics can be considered as the most challenging task [37]. Han et al. focused on reinforcement learning with special attention to multiagent systems.

Most of the reviewed studies in this area use hierarchical central-local agent structures embedded in building models to balance energy consumption and occupant comfort [37].

**Table 7.** Joint conditional probability measure (*jcpm*) of the items learning system and predictive control networked with different building trades and centralized and decentralized system designs, determined by using VOSviewer.

| Item | Learning System (*jcpm*) | | | | Predictive Control (*jcpm*) | | | |
|---|---|---|---|---|---|---|---|---|
| | 2010 and Earlier | 2011–2015 | 2016–2020 | All | 2010 and Earlier | 2011–2015 | 2016–2020 | All |
| lighting control | 0.1370 | 0.1283 | 0.1640 | 0.1372 | 0.0000 | 0.0251 | 0.0000 | 0.0137 |
| daylight control | 0.0000 | 0.0458 | 0.0661 | 0.0426 | 0.0000 | 0.0000 | 0.0000 | 0.0000 |
| hvac control | 0.0000 | 0.1950 | 0.2323 | 0.1784 | 0.0000 | 0.3510 | 0.1357 | 0.2952 |
| multi-objective control | 0.0000 | 0.1375 | 0.3275 | 0.2163 | 0.0000 | 0.0452 | 0.0000 | 0.0263 |
| decentralized control | 0.0000 | 0.0000 | 0.1161 | 0.0312 | 0.0000 | 0.1607 | 0.0000 | 0.1004 |
| centralized control | 0.3000 | 0.0000 | 0.0000 | 0.0895 | 0.0000 | 0.1786 | 0.0000 | 0.0918 |
| user-centered | 0.0000 | 0.0000 | 0.1573 | 0.0600 | 0.0000 | 0.0000 | 0.0982 | 0.0323 |

**Table 8.** Normalized occurrences of building trades and centralized and decentralized system designs as well as the normalized occurrences of the items learning system and predictive control, determined by using VOSviewer.

| Item | Normalized Occurrences | | | |
|---|---|---|---|---|
| | 2010 and Earlier | 2011–2015 | 2016–2020 | All |
| lighting control | 0.1688 | 0.1462 | 0.1935 | 0.1652 |
| daylight control | 0.0500 | 0.0577 | 0.0645 | 0.0574 |
| HVAC control | 0.0063 | 0.0962 | 0.0323 | 0.0539 |
| multi-objective control | 0.0000 | 0.0577 | 0.0839 | 0.0487 |
| decentralized control | 0.0188 | 0.0462 | 0.0323 | 0.0348 |
| centralized control | 0.0125 | 0.0115 | 0.0065 | 0.0104 |
| user-centered | 0.0313 | 0.0308 | 0.0516 | 0.0365 |
| learning system | 0.0625 | 0.1538 | 0.2000 | 0.1409 |
| predictive control | 0.0188 | 0.1615 | 0.0903 | 0.1026 |

In their literature review, Jain and Garg point out that lighting is preferably practised with closed-loop control, as these are mostly geared towards simple energy reduction. Open-loop controls (multi-objective) are mostly simulation-based and include comfort criteria (glare) in addition to energy objectives [8]. In this context, Shen, Hu and Patel compare independent (closed-loop) and integral (open-loop) control strategies, which include daylight, artificial lighting and HVAC [5]. While the independent controls use sensor inputs and rule-based algorithms, the integral control involves co-simulation with EnergyPlus, BCVTB and Matlab. The latter category shows better performance. In addition, Shen, Hu and Patel find that multi-objective control using indoor temperature as input as well as external solar radiation is most effective in balancing comfort and energy demand [5]. Alkhatib et al. describe different control approaches for adaptive façades [33]. The prediction of the system behaviour has to deal with uncertainties due to changing weather conditions, unpredictable user behaviour as well as the influence of the building structure itself. The paper distinguishes between extrinsic (closed-loop) control strategies that use a feedback loop and react to different conditions, including those that were not expected during system design, and intrinsic (open-loop) controls that make a decision based on the environmental conditions and can also integrate other specific influences such as façade specifications or user behaviour into the control strategy.

### 3.5. From Centralisation to Decentralisation

The analyses, based on the VOSviewer results, further show that decentralized control systems are gaining importance and the relevance of centralized controls is declining (see Table 8, column of normalized occurrences). This change is also reflected in connection with learning systems. Early studies of learning systems were related to centralized systems (*jcpm*: 0.3, time period 1, see Table 7). In contrast, the studies from the most

recent publication period list a link to decentralized system architectures (*jcpm*: 0.1161, time period 3).

Some of the reviewed articles analyzed reinforce this trend, by highlighting the importance of individual lighting controls (e.g., occupancy-based artificial lighting control) [8,11,20,22,27,31,36] and especially decentralised control logic [26,37,44]. By using dynamically adjustable lighting scenarios, lighting conditions can be adapted to specific user requirements and preferences to improve comfort. For example, with the integration of illuminance sensors in the luminaire, each luminaire can be customized [22]. In this context, Bellia et al. identify some studies on daylight and occupancy dependent lighting concepts, based on sensor integration [22].

The result is a focus on lighting concepts that move away from room-oriented lighting control to clustered implementations. Whether the underlying control system itself is also to be implemented decentrally or centrally should be determined by the system performance depending on the target application.

Pandharipande and Caicedo discuss design aspects of lighting control systems with luminaire-based sensors. The results list that a distributed control architecture (control intelligence in the luminaire) offers advantages to realise future lighting applications. This is because smart luminaires require a high level of communication capability as well as simple and adaptable programming and implementation [44]. In addition, Al-Ghaili et al. state that ICT-based lighting controls are highly relevant to achieve more efficient lighting systems and to promote the smart building idea [26].

In this context, however, it is necessary to define appropriate sensor networks and to ensure sufficient interoperability and communication stability—especially if several systems are to use the same information, e.g., the use of occupancy information for the daylighting and artificial lighting trades as well as for the HVAC trades. Wireless sensor networks offer certain advantages over wired communication networks due to the low installation effort and higher flexibility [50]. Park et al. also come to a similar conclusion that occupancy-based controls (OCC) require robust data acquisition systems [20].

In their review, Bellia and coworkers make a request for future work that more consideration be given to user behavior in order to define parameters and control algorithms that better ensure dynamic user requirements for visual comfort than static lighting [22]. With the increased interest in individualized lighting, occupancy-based lighting controls need to extract presence information at the zone level [20]. Technological support through data storage and data analysis, communication networks, and especially suitable sensor technologies are therefore considered a necessary prerequisite for the implementation of individual lighting.

### 3.6. The Sensor Resolution as a Key Technology

In addition, the results of the VOSviewer based analysis include that illuminance sensors, like occupancy sensors, were working content of the reviewed articles across all time periods. While the two sensor types predominantly played a minor role in lighting controls and building energy demand in 2010 or before, a stronger linkage emerged in the more recent publication periods. For example, considering occupancy detection, the increase in the *jcpm* with respect to lighting controls is 50.5% (time period 1 to period 3), and with respect to building energy demand, the increase is 22.2% (time period 1 to period 3, see Table 9). While occupancy detection had the highest relevance from 2011 to 2015 (normalized occurrences of occupancy detection: 0.0962, Table 10), the importance of light sensors remained almost the same across all time periods.

**Table 9.** Joint conditional probability measure (*jcpm*) of the items lighting sensors and occupancy detection networked with the items lighting control and building energy savings, determined by using VOSviewer.

| Item | Lighting Sensors (*jcpm*) | | | | Occupancy Detection (*jcpm*) | | | |
|---|---|---|---|---|---|---|---|---|
| | 2010 and Earlier | 2011–2015 | 2016–2020 | All | 2010 and Earlier | 2011–2015 | 2016–2020 | All |
| lighting control | 0.0000 | 0.3395 | 0.2833 | 0.2186 | 0.1919 | 0.2321 | 0.2889 | 0.2292 |
| building energy savings | 0.0000 | 0.1059 | 0.2635 | 0.1230 | 0.4079 | 0.2847 | 0.4985 | 0.3545 |

**Table 10.** Normalized occurrences of lighting sensors, occupancy detection, lighting control and building energy savings, determined by using VOSviewer.

| Item | Normalized Occurrences | | | |
|---|---|---|---|---|
| | 2010 and Earlier | 2011–2015 | 2016–2020 | All |
| lighting control | 0.1688 | 0.1462 | 0.1935 | 0.1652 |
| building energy savings | 0.2438 | 0.3269 | 0.4774 | 0.3443 |
| lighting sensors | 0.0250 | 0.0192 | 0.0258 | 0.0226 |
| occupancy detection | 0.0688 | 0.0962 | 0.0581 | 0.0783 |

This development is also reflected in the study results of Al-Ghaili et al., in which the occupancy-dependent control of lighting systems plays a relevant role in energy savings [26]. This is especially true for office buildings. It was also found that workplace-individual lighting controls are more effective in achieved energy savings in combination with a room-wide or zone-wide control strategy. Individual occupancy-based control strategies could in this way achieve up to 30% in additional lighting energy savings [26]. Dubois et al. also emphasize in their work that the use of electric lighting controls in combination with the right sensor network can contribute significantly to reducing energy consumption [31]. In this context, Gentile, Dubois, and Laike point out the importance of sensor position in daylight-assisted lighting controls [23]. Here, the recommendation is made to differentiate between control with an illuminance sensor facing the interior (measuring the combination of daylight and artificial lighting) and open control with a sensor facing only the incidence of daylight. In addition to sensor resolution and positioning, zoning in particular plays a significant role in occupancy detection in the reviewed papers on energy reduction [8,9,11,22,23,26,27,30,31,33,35,36,45–47].

Salimi and Hammad provide a comprehensive overview of different modeling methods for the detection of occupancy-related parameters that influence the energy demand of a building [36]. Occupancy detection through modeling can be divided at different levels of detail: building level, room level, room level considering occupants, and occupant level. Detailed occupancy monitoring is highly relevant to gain a database and methods are highlighted in the review, including the use of motion sensors, vision-based technologies, radio-frequency-based (RF-based) technologies, multi-sensor networks, virtual occupancy sensors, and surveys and post-occupancy evaluations. A literature review showed that most studies use tracked occupancy data to easily determine occupancy duration. In terms of zoning and spatial resolution, most of the papers reviewed focused on the individual level, but only a small proportion also included specific user preferences. Salimi and Hammad suggest capturing occupant and environmental parameters in near real-time to account for specific user preferences and thus develop and operate appropriate control strategies. In this perspective, Building Information Modeling (BIM) as a digital representation of a building can play a significant role—once by providing all relevant information about installed Internet of Things (IoT) devices in a building (e.g., sensors type and location) and their collected sensory data, but also by including occupancy related information in the future. Thus, BIM has the potential to reduce communication gaps and to help improve the interoperability between different building control systems [36].

## 4. Discussion

From the results examined by the bibliometric review at the level of research articles (see Section 3.1), current trends in control strategies' research can be deduced. The results show clearly that building controls increasingly address not only one goal, but serve multiple requirements in parallel. This becomes clear in the growing interest in multi-target controls in general (see occurrences in Table 5) and especially the mutual interconnection of the items comfort and energy demand under the aspect of multi-target control (see also Table 6). This trend is also consistent with review-level statements. While some previous work tended to focus on energy savings, Ding et al. observe a trend toward integrating visual comfort and energy efficiency in their review article on control logics for office lighting [42]. Since the inclusion of visual comfort criteria in the control logic can be assumed to lead to higher user acceptance, a lower user override can be assumed. As a positive consequence, user-centeredness can result in a better performance in terms of energy than compared to a control system designed exclusively for saving energy. Literature addressing this paradoxical effect is underrepresented within the literature reviewed, so more focus in this direction should be considered for future research efforts. Based on the increasingly strong interest in multi-objective control, it can be concluded that the importance of combining energy efficiency and user comfort is recognized. In the following, the initial research questions are juxtaposed with the findings of the literature study.

### 4.1. Individual Lighting as a Key Technology—RQ1

With RQ1, the overarching question of this review was formulated as to how energy efficiency and user comfort can be taken into account in lighting control strategies in office buildings.

Daylighting can significantly affect the thermal situation in the building. An improvement of the latter is achieved by largely allowing daylight and solar thermal gains to pass the facade during heating periods, preventing only present users from glare, and by blocking excessive daylight and thermal entry during cooling periods providing only present users with an adequate amount of daylight. A live detection of the number and location of occupants in the room must be implemented to realize such a control logic. The fact that daylight entry and solar thermal gains cannot be completely separated makes such efforts necessary, but separation between thermal and visual light transmittance of window systems is currently researched in the field of applied material science. In this respect, switchable glazing makes it possible to control the transmission of light in the visual and thermal frequency range independently to a certain level [51,52], which helps to improve daylight utilization especially during cooling periods. Such a separation in thermal and light transmittance further increases the demands on proper control logics and strategies.

In the authors' opinion, a key feature for future integral control systems is the ability to tailor not only artificial lighting but also daylighting to each user individually, which reduces artificial lighting energy demand to a minimum. Simultaneously, heating and cooling energy use can be reduced to a minimum, which is crucial for thermal comfort. Moreover, the lighting preferences of the individual users can be better addressed, which leads to an enhancement of the visual comfort. This is as well demonstrated by the results in Section 3.3, where it is shown that an implementation of both aspects—a higher comfort and a higher energy efficiency—requires a stronger user-centering, i.e., by an improved mapping of the user behavior in the building in order to make the resulting effects comprehensible.

Simulation-based controls [53] and learning systems [37] are identified in the literature as methods to ensure the required user integration. However, research gaps are currently identified for both systems that stand in the way of greater applicability.

*4.2. Improving System Strengths to Minimize System Weaknesses—RQ2*

In addition to the fact, as described above in the answer to RQ1, that an increasing trend at research level towards the establishment of multi-objective control could be identified through numerous literature sources, RQ2 attempted to critically address the different strengths and weaknesses of the systems and concepts used, which could accelerate or hinder real implementation. In this context, Jain and Garg state in their review article that there is a lack of simulation tools that are fast and user-friendly enough to be integrated into control processes [8]. The development of fast and reliable techniques to build a simulation-based control routine is identified as a challenge here. Limitations lie especially in less accurate real-time estimations provided by simulations (e.g., sky state or discomfort glare) and running times, which inhibits the application in practice clearly.

Konstantoglou and Tsangrassoulis and Tabadkani et al. note in their reviews that user preferences are hardly taken into account in lighting control logics [9,38]. This fact is also reflected in the relatively low binding between the items 'user-centered' and 'learning systems' ($jcpm$: 0.06, period: All) and between 'user-centered' and 'predictive control' ($jcpm$: 0.0323, period: All, see also Table 7). Especially for the latter, the connection to the item user-centered is only very slightly developed. This fact is also confirmed by Rockett and Hathway. In their review of model predictive control studies, a deficit of user intervention options could be identified [32]. Greater consideration of individual preferences in control logic promises less system intervention in energy-efficiently oriented automated control curves [23]. In summary, taking user preferences more into account may also improve energy efficiency. In order to increase user satisfaction, applied studies in real environments, e.g., living labs, should also be enhanced in the future. This can help to gain important insights through post-occupancy evaluations as well as direct user feedback from surveys [11].

In the VOSviewer based analysis, an increasing correlation was found between the items 'occupancy detection' and 'lighting control' (see also Table 9). Improving occupancy detection for control systems is shown to be beneficial for energy efficiency because artificial lighting can be reduced in areas where it is not needed [54]. In this context, Athienitis and Tzempelikos as well as Ryckaert et al. point out that a reduction in artificial lighting use is also associated with a reduction in cooling demand [55,56]. The authors state that, in the case of daylight control, beneficial energy effects can be expected by including detailed occupancy detection. Furthermore, considering a winter situation, heating energy demand can be reduced by opening the shading system on large parts of the facade, while only single shading elements are closed to protect each occupant from glare individually, depending on the sitting position in the room and the incident sun angle. In addition, in the summer situation, opening only single shading elements to facilitate sufficient daylight for individually occupied workplaces, while closing the rest of the facade, can significantly help to avoid overheating. Literature studies on this topic proved to be underrepresented in the literature reviewed, so this should be the subject of research in future work.

In summary, individual lighting controls can offer the potential to reduce energy demand and increase comfort through stronger user integration (consideration of preferences) and user mapping (information on presence and user behavior). Current weaknesses can mainly be found in the implementation of user integration and user mapping. This can be counteracted by higher sensor integration and sensor networks. The result is that further improvements in comfort and energy efficiency can be achieved through improved user-centeredness. It should be noted that the energy requirements of the system participants for user mapping do not outweigh the positive effects.

*4.3. Integral Concepts and Interoperability as a Future Requirement of Control Technology—RQ3*

In this perspective, RQ3 claims to answer, based on the literature review conducted, whether and how current and future research trends can meet the needs of future control strategies. The findings presented in Section 3.4 show that, although primarily synergies between daylight and artificial light can be perfectly matched within a control system, this

is not widely applied. For example, the VOSviewer based analysis results show only a slight consideration of daylight utilization in the learning system (*jcpm*: 0.0426, period: All) and none at all in the predictive control (*jcpm*: 0, period: All, see also Table 7). The low level of interconnectedness of daylighting to learning and predictive systems is particularly evident from the fact that the interconnectedness *jcpm* of artificial lighting to the two control systems is significantly larger (Table 7). The interconnectedness of the HVAC trade to learning and predictive control systems is also much more pronounced than is the case for the daylighting trade. However, a holistic system approach could lead to a significant reduction in energy demand of 30–80% in artificial lighting energy [4–6] and 3–43% in HVAC energy demand [7].

A larger proportion of glazing does not necessarily lead to a reduction in lighting energy requirements. Higher daylight levels raise the risk of glare, which leads to the need for shading and, as a consequence, artificial light is switched on [57]. Consequently, the energy demand increases. In addition, such manual translations are usually not immediately reset after the potential glare effect is exceeded, which increases the negative effect on the energy demand. Furthermore, higher daylight incidence is prone to higher solar gains. In this context, the additional heat must be compensated by energy-consuming cooling. The thermal effects of daylight must be taken into account accordingly [41,58]. It is therefore highly recommended to apply an integral approach. Integrated daylight and artificial lighting controls offer the potential to reduce energy requirements and improve visual comfort [27]. Better performance could be achieved by including HVAC controls in the integral system [5]. With the identified influence of occupancy information on energy demand, it follows that there is not only a need for a generally applicable integral system approach, but also for an integral system approach that takes the occupancy situation into account [20].

The focus on integral concepts can thus be identified as a future requirement for control systems. An integral approach can not only offer a reduction in energy demand, but also improvements in comfort and user satisfaction, as these can be understood as the sum of a user assessment of the overall situation. A cross-trade information provision of sensor-collected data offers the possibility to increase the efficiency of integral concepts. Interoperability between the systems must therefore be a prerequisite.

## 5. Conclusions

The growing importance of proper controls of daylight- and artificial lighting systems, which allow optimizing a building's energy efficiency while considering visual and thermal comfort is underpinned by the results of the reviewed literature. User acceptance is mentioned in the literature as a decisive factor for achieving targeted system efficiencies. User acceptance of the control system is largely determined by the ability of the user to interact with the control system. Since a high degree of automation is in principle associated with higher energy savings than with partially automated and manual systems, and since system interventions, as a result of insufficient user acceptance, can influence the performance of automated systems noticeably, it is essential to take greater account of user behavior. Although the literature analysis identifies some centralized systems that strive for individualized control, the trend is moving towards decentralized concepts—in particular by means of occupant detection via sensor technology as well as appropriate clustering and zoning, which both prove to be necessary requirements for the implementation of individualized solutions and thus count as the most important factors for achieving further energy efficiency increases and comfort improvements.

In the author's opinion, taking various trade-specific influences into account in integral control concepts is key to reducing buildings' energy demand. User preferences and user behavior should not be neglected in the course of merging the trades into an overall building assessment because users have a decisive influence on performance and acceptance of the system. If acceptance by the user is not guaranteed, this can have a negative impact on the applicability and spread of such control systems. Intelligent commissioning

can also help to increase user acceptance. Smart commissioning supports the reduction of complexity during system commissioning, which can reduce the risk of faulty operation. Besides the ability to provide a database of information by embedded devices and gadgets, BIM can also play a significant role in the near future by incorporating all relevant information from the involved trades and infrastructure to fully setup and configure a multi-objective control routine. In this way, it builds the ideal platform to realize a smart commissioning, reduce malfunctions and furthermore enable comprehensive monitoring during operation.

**Supplementary Materials:** The following are available online at https://www.mdpi.com/1996-1073/14/13/3852/s1.

**Author Contributions:** conceptualization, D.P., S.H., M.H., and V.v.K.; methodology, D.P., S.H., M.H., and V.v.K.; formal analysis—jcpm evaluation, D.P. and S.H.; formal analysis—review evaluation, M.H. and V.v.K.; data curation—literature survey, D.P., S.H., M.H., and V.v.K.; writing—original draft preparation, D.P., S.H., M.H., and V.v.K.; writing—review and editing, D.P., S.H., M.H., and V.v.K.; visualization, D.P. and S.H.; supervision, R.P.; project administration, D.P.; funding acquisition, R.P., M.H. and D.P. All authors have read and agreed to the published version of the manuscript.

**Funding:** The implementation of this study took place within the framework of the research project Bim2IndiLight (https://www.uibk.ac.at/bauphysik/forschung/projects/bim2indilight/index.html.en (accessed on 25 June 2021)), which is funded by the Province of Tyrol and the European Regional Development Fund (EFRE) under the grant agreement number EFRE K-Regio 10033.

**Institutional Review Board Statement:** Not applicable.

**Informed Consent Statement:** Not applicable.

**Data Availability Statement:** The research papers used for the VOSviewer work and the jcpm analysis as well as the assignment of the found key words by VOSviewer to the defined items (see Section 2.2) can be downloaded as PDF at the Supplementary Materials: https://www.mdpi.com/1996-1073/14/13/3852/s1 (accessed on 25 June 2021).

**Acknowledgments:** The publication of this review article was supported by the Unit of Energy Efficient Buildings at the University of Innsbruck and by Bartenbach.

**Conflicts of Interest:** The authors declare no conflict of interest.

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
