# Peer review of "Control Strategies for Daylight and Artificial Lighting in Office Buildings—A Bibliometrically Assisted Review"

_energies, doi:10.3390/en14133852_

Round 1

Reviewer 1 Report

The Abstract of this paper is saying that goal of it is to provide an overview of the recently published literature on enhanced control strategies, in which new control approaches are critically analysed regarding the fulfilment of energy efficiency targets and comfort criteria simultaneously. But "the recently papers" are not enough to make this kind of review. And what does means recently - from last week, last year or 10 years? The literature review must be complited and done on the papers which  are "mil-stones" for that subject. In this review there is not enought number of the papers which are dealing with technical solutions for achiving daylighting control at energy-efficient lighting systems. This review paper does not contain enough technical and technological information derived from papers  like:

  1. G. Boscarino and M. Moallem, "Daylighting Control and Simulation for LED-Based Energy-Efficient Lighting Systems," in IEEE Transactions on Industrial Informatics, vol. 12, no. 1, pp. 301-309, Feb. 2016, doi: 10.1109/TII.2015.2509423.
  2. https://doi.org/10.1016/j.buildenv.2008.04.016
  4. https://www.researchgate.net/profile/Larry-Degelman/publication/242719069_A_Model_for_simulation_of_daylighting_and_occupancy_sensors_as_an_energy_control_strategy_for_office_buildings/links/0c9605315007c60dcd000000/A-Model-for-simulation-of-daylighting-and-occupancy-sensors-as-an-energy-control-strategy-for-office-buildings.pdf

The authors of the paper under review dose not must to cite the particular papers listed above but must extended their literature review to the key papers for the subject (does not metter how fare ago those papers were published).

Reviewer 2 Report

  1. Line 43: What does HVAC stand for? All abbreviations must be deciphered.
  2. Lines 62-63: “How can energy efficiency and user comfort be addressed in lighting control strategies in office buildings?” In universities students learn courses concerning optimal lightning. There are standards for lightning. The authors should rewrite objective of this study.
  3. Lines 67-68: “In doing so, current obstacles of existing control approaches and resulting research gaps will be identified for future investigations in this area.” This is unacceptable. Research gaps must be revealed before the objective of the article.
  4. There are some mistakes. For example, line 137 “Therefor”.
  5. Lines 137-139: “Therefor the selected research articles were first divided in three time-periods as listed in Table 3: 2010 or earlier, from 2011 to 2015 and from 2016 to 2020.” The authors should explain this division.
  6. Tables must be formed according to instructions for authors.
  7. Lines 570-571: “The results of the reviewed literature underpin the growing importance of control strategies to optimize a building’s energy efficiency …..”. This statement is trivial.
  8. Lines 586-587: “In the authors opinion, integral control concepts offer the possibility to significantly reduce the energy demand ….” This statement is trivial.
  9. This research does not contain actual information about lighting such as lighting standards, lighting costs, lighting control system cost, etc. Without this, the article is just empty.

Reviewer 3 Report

  • Is necessary to explain the control method available nowadays and the future development, to know de different possibilities of regulation.
  • We have different lights sources, explain the advantages and disadvantages of each one.
  • If not propose changes in illumination method in an existent building, explain de regulation methods applied in this cases,
  • In a new building what is the criteria to select a control method and the light source.
  • Compare different sources and control methods to see the solutions propose by the authors.
  • Result section is not right organized. Is difficult to read, please redone.

Round 2

Reviewer 1 Report

The paper is improved. It is worth to publish it. 

Reviewer 2 Report

  1. Tables 2, 4, 5, 6 must be formed according to instructions for authors.
  2. This research does not contain actual information about lighting such as lighting standards, lighting costs, lighting control system cost, etc. Without this, the article is just empty.

Reviewer 3 Report

The response of authors is not properly. Is neccesary to respond in rigth way the questions.

Major revision